# Single-cell RNA sequencing reveals a heterogeneous response to Glucocorticoids in breast cancer cells

Jackson A. Hoffman [1], Brian N. Papas[2], Kevin W. Trotter[1] & Trevor K. Archer[1✉]

Steroid hormone receptors such as the Glucocorticoid Receptor (GR) mediate transcriptional responses to hormones and are frequently targeted in the treatment of human diseases. Experiments using bulk populations of cells have provided a detailed picture of the global transcriptional hormone response but are unable to interrogate cell-to-cell transcriptional heterogeneity. To examine the glucocorticoid response in individual cells, we performed single cell RNA sequencing (scRNAseq) in a human breast cancer cell line. The transcriptional response to hormone was robustly detected in individual cells and scRNAseq provided additional statistical power to identify over 100 GR-regulated genes that were not detected in bulk RNAseq. scRNAseq revealed striking cell-to-cell variability in the hormone response. On average, individual hormone-treated cells showed a response at only 30% of the total set of GR target genes. Understanding the basis of this heterogeneity will be critical for the development of more precise models of steroid hormone signaling.

---

[1] Chromatin and Gene Expression Section, Epigenetics and Stem Cell Biology Laboratory, National Institute of Environmental Health Sciences, Research Triangle Park, Durham, NC, USA. [2] Integrative Bioinformatics, Epigenetics and Stem Cell Biology Laboratory, National Institute of Environmental Health Sciences, Research Triangle Park, Durham, NC, USA. ✉email: trevor.archer@nih.gov

Cellular heterogeneity is a feature of metazoan evolution. Multicellularity allows for cellular specification, the formation of distinct cell types, and the development of complex body plans. Furthermore, within the context of the body plan, organs and tissues are comprises multiple cell types, and the organization of these heterogeneous cell populations yields diverse functions. Underlying this heterogeneity are differences in the transcriptional programs of each cell. For instance, scRNAseq revealed 39 transcriptionally distinct cell populations within the mouse retina[1], demonstrating the cellular and transcriptional complexity of multicellular life.

Cellular heterogeneity is also observed within individual cell types. Cell-to-cell variability is observed in the transcriptomic profiles of individual cells within each of the 39 cell types observed in the mouse retina[1]. Such variability has been observed within diverse cell types across hundreds of scRNAseq publications. Intriguingly, heterogeneity is even observed amongst clonal populations of single-cell organisms such as *E. coli*[2] and *S. cerevisiae*[3]. Similarly, significant cellular heterogeneity is observed between individual cells in human tumors[4–6], which poses a major obstacle to the profiling and treatment of human cancers. For instance, the characterization and treatment of breast cancers is often dependent on the expression profiles of hormone receptors. However, the proportion of cells expressing each hormone receptor within an individual tumor varies wildly[7–9] hindering treatment decision and patient prognoses.

Beyond the varied transcriptional profiles and patterns of hormone receptor expression that are observed within individual cell types or tumors, experiments performed with reporter genes have suggested that there is also heterogeneity in the cellular response to hormone. In clonal populations of mouse Ltk-cells harboring low copy numbers of an MMTV-driven LacZ reporter, induction of LacZ expression by the synthetic glucocorticoid dexamethasone (Dex) was dose-dependent and significant heterogeneity was observed between cells within individual colonies[10]. Cellular heterogeneity in the Dex response was also observed in a flow cytometry-based reporter assay in Cos-1 African green monkey kidney fibroblasts and HTC rat hepatoma cells[11]. Cell-to-cell variability was also demonstrated at the RNA level by performing RNA-FISH for MMTV transcripts in the 3617 mouse mammary carcinoma cell line[12]. Finally, live imaging of transcription from an integrated MMTV-reporter in response to glucocorticoids[13] or a modified endogenous gene in response to estrogen[14] revealed extensive cell-to-cell heterogeneity in living cells. Thus, the transcriptional response to hormone at individual genes exhibited considerable cellular heterogeneity.

In this study, we sought to interrogate the cellular heterogeneity underlying the global transcriptional response to glucocorticoids in human breast cancer cells. Recently published work has demonstrated that the Dex response can be observed using scRNAseq[15]. Here, we utilize scRNAseq to detect a robust transcriptional response across an 18 h time course of Dex treatment. Comparison to bulk RNAseq revealed that scRNAseq was biased toward detection of highly expressed genes and was thus limited in the ability to detect GR target genes. However, hundreds of GR target genes exhibited a response in single cells. Over 100 of these detected GR target genes were not detected in bulk RNAseq, demonstrating that scRNAseq could provide novel insight into the types of genes regulated by GR. Finally, scRNAseq revealed significant cellular heterogeneity within the Dex response of endogenous GR target genes. While a response was detected in every Dex-treated cell, individual cells on average showed a response at only 30% of detected GR-regulated genes. Taken together, these data demonstrate that the transcriptional response to hormone is highly variable in individual cells.

## Results

**Detecting the Dex response in single cells.** To examine the transcriptional response to Dex in single cells, we utilized the T47D A1–2 human breast cancer cell line that we have previously used to characterize GR-chromatin interactions[16]. We performed both scRNAseq and bulk RNAseq using A1–2 cells that were treated with 100 nM Dex for 1, 2, 4, 8, and 18 h timepoints or with ethanol vehicle for 18 h. Cells for all timepoints were collected together, and vehicle-treated control cells will be referred to as "EtOH-treated cells" or "0 h" cells in the following text and figures. After data processing (see materials and methods), the final scRNAseq dataset used for analysis featured 400 cells per treatment timepoint with an average of 29,448 unique transcripts per cell and 6415 genes per cell (Fig. 1a, b). The scRNAseq data provided a robust profile of gene expression in A1–2 cells. More than half of the genes detected by bulk RNAseq (including ncRNAs and other non-protein-coding genes) were detected in the scRNAseq dataset, (~56%, Fig. 1c). However, the scRNAseq was biased toward the detection of highly expressed genes, such that when we distributed the genes detected in the bulk RNAseq dataset into quintiles based on their expression levels, nearly all genes in the top quintile and >80% of the genes in the second quintile were detected in the scRNAseq (Fig. 1d). Conversely, fewer than half of the third quintile and approximately a quarter of the fourth and fifth quintile genes are detected in the scRNAseq (Fig. 1d). This bias was reinforced by examining the number of cells in which each gene was detected. The mean detection rate per individual cell was >50% for genes in the top quintile, but <10% for all other genes (Fig. 1d).

We then examined several well established Dex/GR target genes to use as initial candidates to evaluate the Dex response with the two datasets. Based on bulk RNAseq average read counts for EtOH control treated cells, we chose genes that were initially expressed at low, moderate, and high levels in untreated A1–2 cells and showed similar, progressive patterns of induction across the time course of Dex treatment (Fig. 1e). As expected, these genes were differentially detected in the scRNAseq. *Phenylethanolamine N-methyltransferase* (*PNMT*, bulk EtOH mean = 208) was detected in 19% of untreated cells but was detected in greater proportions of cells and expressed at progressively higher levels over the time course of Dex treatment (Fig. 1f). *Glucocorticoid-Induced Leucine Zipper* (*TSC22D3*, bulk EtOH mean = 5670) was detected in 48% of untreated cells, but at the 1 h Dex timepoint it was detected in a majority of cells and both detection and expression progressively increased over subsequent timepoints (Fig. 1g). Finally, *Glutamate-Ammonia Ligase* (*GLUL*, bulk EtOH mean = 41,382) was detected in 93% of untreated cells and expression progressively increased over the first 8 h of Dex treatment (Fig. 1h). Repressed genes were also broadly detected across the dataset, with detection rates decreasing over the Dex time course as the levels of RNA expression decreased (Supplemental Fig. 1). Thus, the scRNAseq replicates the bulk RNAseq transcriptional response for a number of validated Dex/GR target genes.

**scRNAseq reveals targets of glucocorticoid signaling.** We next used the both datasets to call differentially expressed genes (DEGs) for pairwise comparisons of each Dex treatment timepoint and EtOH-treated cells. For the bulk RNAseq data, DEGs were called using DESeq with a fold change cutoff of 1.5 and adjusted *p*-value of 0.05. We detected 172 DEGs at 1 h, 735 at 2 h, 1270 at 4 h, 2056 at 8 h, and 2417 at 18 h (Supplemental Fig. 2a). As the 400 cells per timepoint in the scRNAseq data essentially represent 400 replicates, we reasoned that the fold change cutoff could be relaxed and that we could use a more stringent adjusted

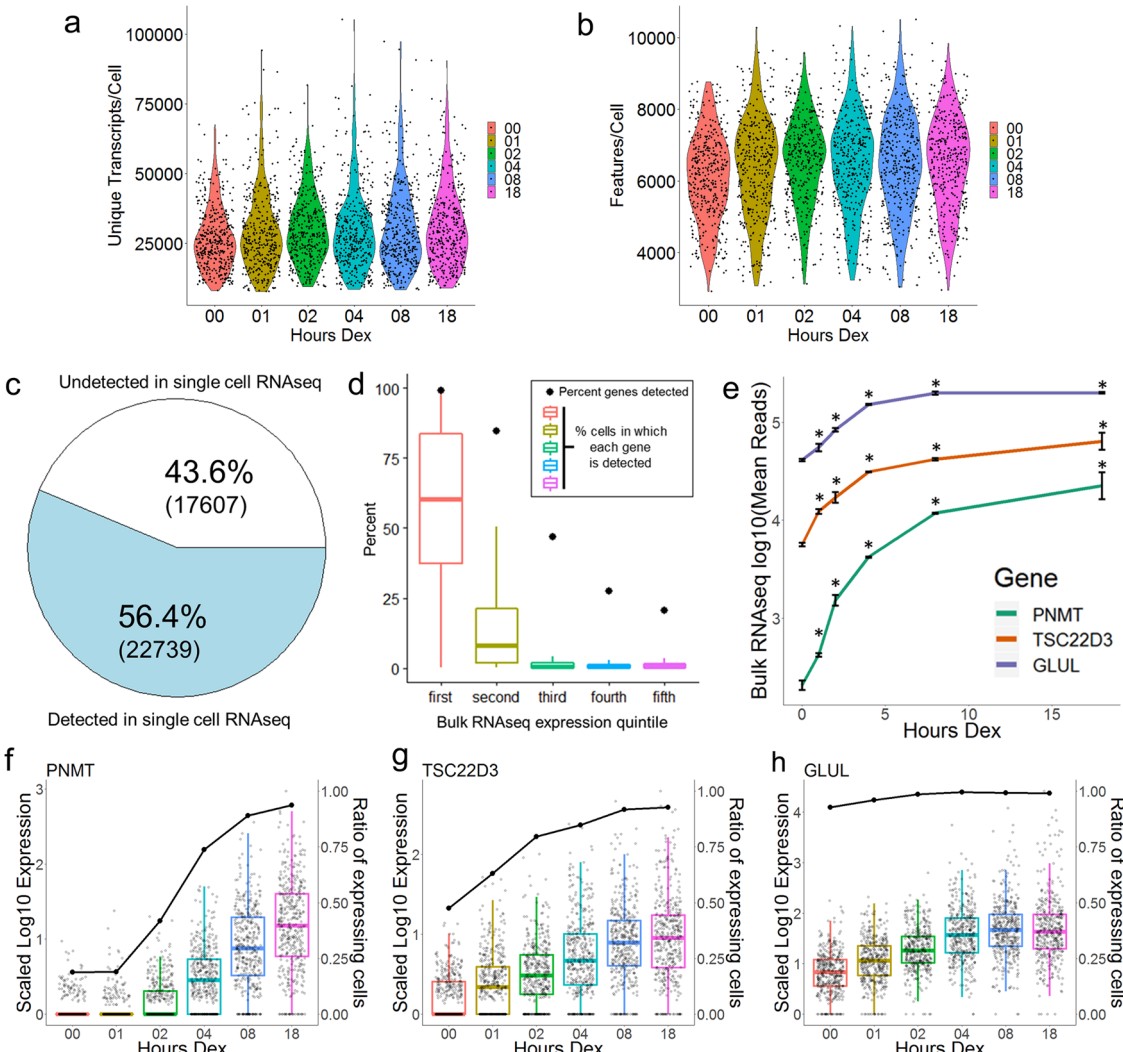

**Fig. 1 Detecting the Dex response in single cells. a, b** Violin plot of unique transcripts per cell and detected genes per cell in scRNAseq dataset. Each dot represents a single cell. Four-hundred cells were examined at each timepoint. **c** Pie chart depicting all genes/genomic features expressed in bulk RNAseq and the proportion detected by scRNAseq. **d** Plot depicting the percentage of genes from each bulk RNAseq expression quintile that are detected in scRNAseq (black dots) and box + whiskers depicting the range of detection rates for each gene in the scRNAseq. **e** Expression patterns for PNMT, TSC22D3, and GLUL across the Dex time course in the bulk RNAseq. Asterisks = *p*-value < 0.05. **f–h** Scaled Log10 expression (box and scatter plots) and ratio of expressing cells (black dot/line plot) for PNMT, TSC22D3, and GLUL in scRNAseq. Each dot in the scatter plot represents an individual cell.

*p*-value cutoff. We used the MAST test with a fold change cutoff of 1.25, adjusted *p*-value of 0.01, and excluding genes detected in fewer than 10% of cells. Far fewer DEGs were called in the scRNAseq, with 77 DEGs at 1 h Dex, 94 at 2 h, 167 at 4 h, 275 at 8 h, and 285 at 18 h (Supplemental Fig. 2b). The bulk RNAseq DEGs that were not called in the scRNAseq (bulk unique DEGs) were expressed at significantly lower levels than shared DEGs and were detected in a smaller fraction of cells in the scRNAseq (Supplemental Fig. 2c, d). Thus, the failure to identify these DEGs in the scRNAseq could be attributed to low levels of expression and detection.

Surprisingly, the DEGs that were called in the scRNAseq dataset were not entirely overlapping with bulk RNAseq DEGs. At each timepoint, a large number of scRNAseq DEGs were not shared with bulk RNAseq, suggesting that a previously unidentified set of Dex/GR target genes were identified by scRNAseq. To identify scRNAseq-unique DEGs and eliminate genes that were simply called at different timepoints, the DEG lists from each timepoint were merged and the superset of scRNAseq DEGs (391 DEGs) was compared to the superset of bulk RNAseq DEGs

(3285). Overall, 139 DEGs were unique to the scRNAseq dataset, and 252 were shared with the bulk RNAseq (Fig. 2a). These 139 scRNAseq-only DEGs included genes that were progressively upregulated, genes that were induced at early timepoints, and genes that were repressed by Dex treatment (Fig. 2b). In general, the scRNAseq-only DEGs had high detection rates in the scRNAseq, and small fold changes in the bulk RNAseq (Fig. 2c and Supplemental Fig. 2d, e). However, a subset of the scRNAseq-only DEGs did appear to have significant fold changes in the bulk RNAseq. Despite having absolute fold changes >1.5, these genes were not significant in the bulk RNAseq due to false-discovery rates >0.05. This was likely due to these genes having high variability between bulk RNAseq replicates. Taken together, these data suggested that the scRNAseq provides additional statistical power to identify DEGs where Dex treatment has a milder effect on gene expression or where the hormone effect is more variable.

To determine what subset of these scRNAseq-unique DEGs were bona fide transcriptional targets of GR, we interrogated our bulk GR ChIP-seq data[16] to determine whether any of these genes were near GR-binding sites in the genome. Analyses from our

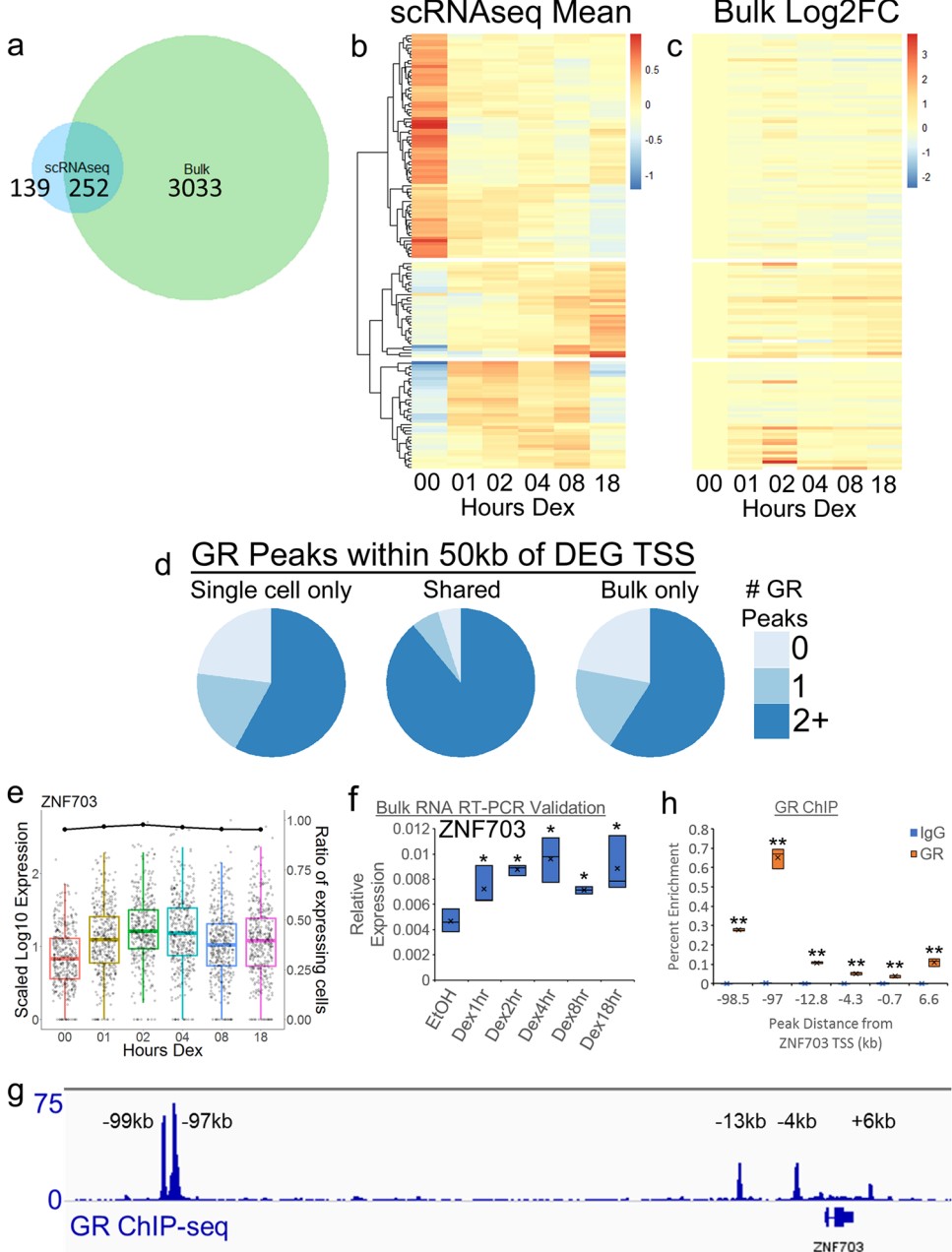

**Fig. 2 scRNAseq reveals targets of glucocorticoid signaling. a** Venn-diagram depicting overlap of DEG supersets from scRNAseq and bulk RNAseq.
**b** Heatmap depicting the mean log-scaled expression for scRNAseq-unique DEGs in the scRNAseq. Heatmap is split into three groups based on the pattern of the Dex response; top = downregulated DEGs, middle = progressively upregulated DEGs, bottom = early induced DEGs. **c** Heatmap depicting Log2FC for scRNAseq-unique DEGs in the bulk RNAseq. Heatmap is in the same order and groups as in **b**. **d** Pie charts showing the proportions of each gene group from **a** that have 1 or 2+ GR peaks within 50 kb of their annotated TSSs. **e** Scaled Log10 expression (box and scatter plots) and ratio of expressing cells (black dot/line plot) for ZNF703 in scRNAseq. **f** Independent RT-QPCR validation of ZNF703 expression in response to Dex. Box plot represents ZNF703 expression relative to the geometric mean of GAPDH, ACTB, and TUBA1B. Horizontal edges and lines in boxes represent three biological replicates and "x" represents the mean * = p-value < 0.05 compared to EtOH. **g** Browser shot of GR ChIP-seq coverage at the ZNF703 locus. **h** GR ChIP-QPCR validation of GR peaks around the ZNF703 locus. Horizontal edges and lines in boxes represent three biological replicates and "x" represents the mean and ** = p-value < 0.01 compared to corresponding IgG enrichment.

previous study demonstrated that most DEGs had GR-binding sites or peaks that were proximal (within 50 kb) to their transcriptional start sites (TSSs)[16]. As such, 78% of bulk RNAseq-specific DEGs had one or more proximal GR peaks (Fig. 2d). The 139 scRNAseq-only DEGs had nearly identical proportions of genes with one or more proximal GR peaks. (Fig. 2d). Intriguingly, the 252 shared DEGs had a much higher proportion of genes with multiple proximal peaks, 95%,

suggesting that the shared DEGs were enriched for genes that were highly bound by GR (Fig. 2d). Overall, the proximity of GR peaks to the TSSs of these DEGs provided strong evidence that they are indeed direct GR target genes. Furthermore, this suggests that scRNAseq identified new targets of Dex signaling.

Of particular interest among these scRNAseq-only DEGs was the zinc-finger protein *ZNF703*. *ZNF703* has previously been identified as an oncogenic driver within the 8p12 amplification

that commonly occurs in estrogen receptor (ER) positive, luminal breast cancers[17–19]. These studies revealed several roles for *ZNF703*, tantamount among these being the suggestion that ZNF703 forms a negative feedback loop with ER in which ER activates ZNF703 expression and ZNF703 in turn represses ER[18]. In the scRNAseq data, ZNF703 was detected in >95% of cells and was induced by Dex across all timepoints, peaking with 2.4-fold upregulation at 4 h (Fig. 2e). Independent validation of ZNF703 expression by reverse transcription PCR (RT-PCR) using bulk RNA revealed an expression pattern similar to that observed in scRNAseq, however, there was considerable variability in ZNF703 expression between biological replicates (Fig. 2f, error bars represent biological triplicates). Similarly, ZNF703 appeared to be Dex-responsive in the bulk RNAseq by fold change with a similar pattern to scRNAseq (Supplemental Fig. 3a), however, it did not pass the adjusted *p*-value significance threshold (<0.05) and was not called as a DEG. This was presumably also due to a high level of variability in ZNF703 expression between biological replicates. As such, scRNAseq uniquely identified *ZNF703* as a Dex-regulated gene.

Analysis of ChIP-seq data revealed multiple GR-binding sites surrounding the *ZNF703* genomic locus. Five GR peaks were found within 15 kb of the *ZNF703* TSS, and an additional eight GR peaks are called in a region between 95 and 175 kb upstream of the TSS that was also occupied by several potential LincRNAs (Fig. 2g and Supplemental Fig. 3b). Independent ChIP-quantitative PCRs (qPCRs) were performed for six of these peaks, and the levels of enrichment strongly correlated with the ChIP-seq data (Fig. 2h). Analysis of previously published ChIP-seq data[16] revealed a Dex-induced increase in K27ac enrichment around the −97kb and −98.5 kb peaks, as well as a recruitment of BRG1 at the −13 kb, −97 kb, and −99 kb peaks (Supplemental Fig. 3c, d). This suggests that these peaks represented potential Dex-dependent enhancers for *ZNF703* expression. Furthermore, using Start-seq data[20], Dex-induced bi-directional enhancer RNA transcription was observed at the GR peaks at −13 kb, −97 kb, and −99 kb (Supplemental Fig. 2c, d). Taken together, these data demonstrated that *ZNF703*, a target gene uniquely identified by scRNAseq, is likely directly regulated by GR through the interaction with multiple potential enhancer sites in response to Dex.

**Dimensional reduction captures the transcriptional response to Dex**. To visualize transcriptional variability over the complete scRNAseq dataset, we performed principal component analysis (PCA) using the top 500 variable genes in the dataset. The Dex response was well represented on the first principal component, with untreated cells clustered on the left side of the PCA plot, and the clusters for each subsequent Dex timepoint positioned to the right of the untreated cluster (Fig. 3a, b). Additional dimensional reduction of the significant principal components was performed using t-distributed Stochastic Neighbor Embedding (t-SNE)[21] and Uniform Manifold Approximation and Projection (UMAP)[22] (Fig. 3c, d and Supplemental Fig. 4a, b). Both methods accentuated the differences between the timepoints of Dex treatment. The t-SNE reduction in particular provided excellent separation between the treatment timepoints, with the timepoints organized in order from the top right to the bottom left of the plot.

To determine whether dimensional reduction would reveal unique cell groupings with similar transcriptional profiles, we utilized the PCA to call de novo cell clusters. When visualized on the PCA and tSNE plots (Fig. 3e, f), the de novo clusters were distinct from the treatment timepoints (Fig. 3b, d). Cluster 5 largely represented EtOH-treated cells, indicating that the majority of untreated cells were transcriptionally similar (Fig. 3g).

Similarly, clusters 4 and 3 were primarily comprised of cells treated with Dex for 18 and 8 h, respectively (Fig. 3g). Conversely, clusters 1, 2, and 6 were comprised of similar numbers of cells from all 6 timepoints (Fig. 3g). Indeed, each cluster contained cells from multiple timepoints, indicating that subsets of cells are transcriptionally indistinguishable across the time course of Dex treatment (Fig. 3g). Thus, while dimensional reduction visually recapitulated the hormone time course, the de novo clusters demonstrated that there was heterogeneity in the transcriptional response to Dex.

**Cellular variability within the hormone response**. We also observed considerable cell-to-cell variability in the degree of hormone responsiveness at individual genes. At any given gene, there were Dex-treated cells that exhibited expression similar to the mean level of expression in EtOH-treated cells (for example, see Fig. 1f–h). This was observed at all Dex-responsive genes, including *FK506 Binding Protein 5* (*FKBP5*), the most robustly responsive gene in the dataset. At 8 and 18 h of Dex treatment, FKBP5 was detected in every cell and was respectively 40-fold and 35-fold upregulated. However, among the 800 cells examined at these timepoints, there was considerable variability in the level of upregulation relative to EtOH-treated cells. As it was not possible to measure FKBP5 levels both before and after Dex treatment in single cells, we sought to determine whether individual cells expressed FKBP5 above the normal range of expression levels in EtOH-treated cells. To do this and determine whether *FKBP5* showed a Dex response in each cell, we determined the ratio of Dex-treated cells that expressed FKBP5 at levels greater than one standard deviation (SD) above the mean level of expression in EtOH-treated cells. Using this threshold, 98% of 8 and 18 h Dex-treated cells showed a response at *FKBP5*, but 14 cells at these timepoints failed to upregulate FKBP5 above the SD of expression in EtOH-treated cells (Fig. 4a). Genes that were repressed by Dex treatment, such as *IER3*, showed similar variability. Using the ratio of cells that were more than one SD *below* the mean level of expression in EtOH-treated cells as a threshold, IER3 was downregulated in 72% of 4 and 8 h Dex-treated cells (Fig. 4b). Thus, there was heterogeneity in the extent to which Dex treatment effected the expression of both induced and repressed target genes in individual cells.

To further characterize the cell-to-cell heterogeneity in the Dex-response, we sought to determine how many Dex target genes showed a response in each cell. To calculate a "Ratio of Responding Genes" (RRG), the mean log-scaled expression level and standard deviation was determined for each DEG in untreated cells. To avoid underestimating the baseline level of gene expression (and thus over-estimating the Dex response), undetected/zero values were excluded from this calculation. As above, we used the SD of expression levels in EtOH-treated cells to set thresholds. We called a gene "responsive" in each cell if it was expressed at a level greater than one SD above the mean EtOH-treated level (or more than one SD below the mean for downregulated genes). We performed this calculation for all scRNAseq DEGs to determine the RRG in each cell. As cell-to-cell variability was present in EtOH-treated cells and this calculation was based on SD, some Dex target genes were called as responding in EtOH-treated cells. On average, EtOH-treated cells had a 12% RRG, indicating the baseline variability in A1–2 cells. Upon 1 h Dex treatment, the RRG nearly doubled to 22% (Fig. 4c). The RRG increased with prolonged Dex treatment to a maximum of 37% at 18 h (Fig. 4c). Among all timepoints, the most responsive cells had RRGs between 50% and 65% (Fig. 4c). When taking the 2000 Dex-treated cells across all timepoints together, the mean RRG was ~30%. Thus, the average Dex-treated

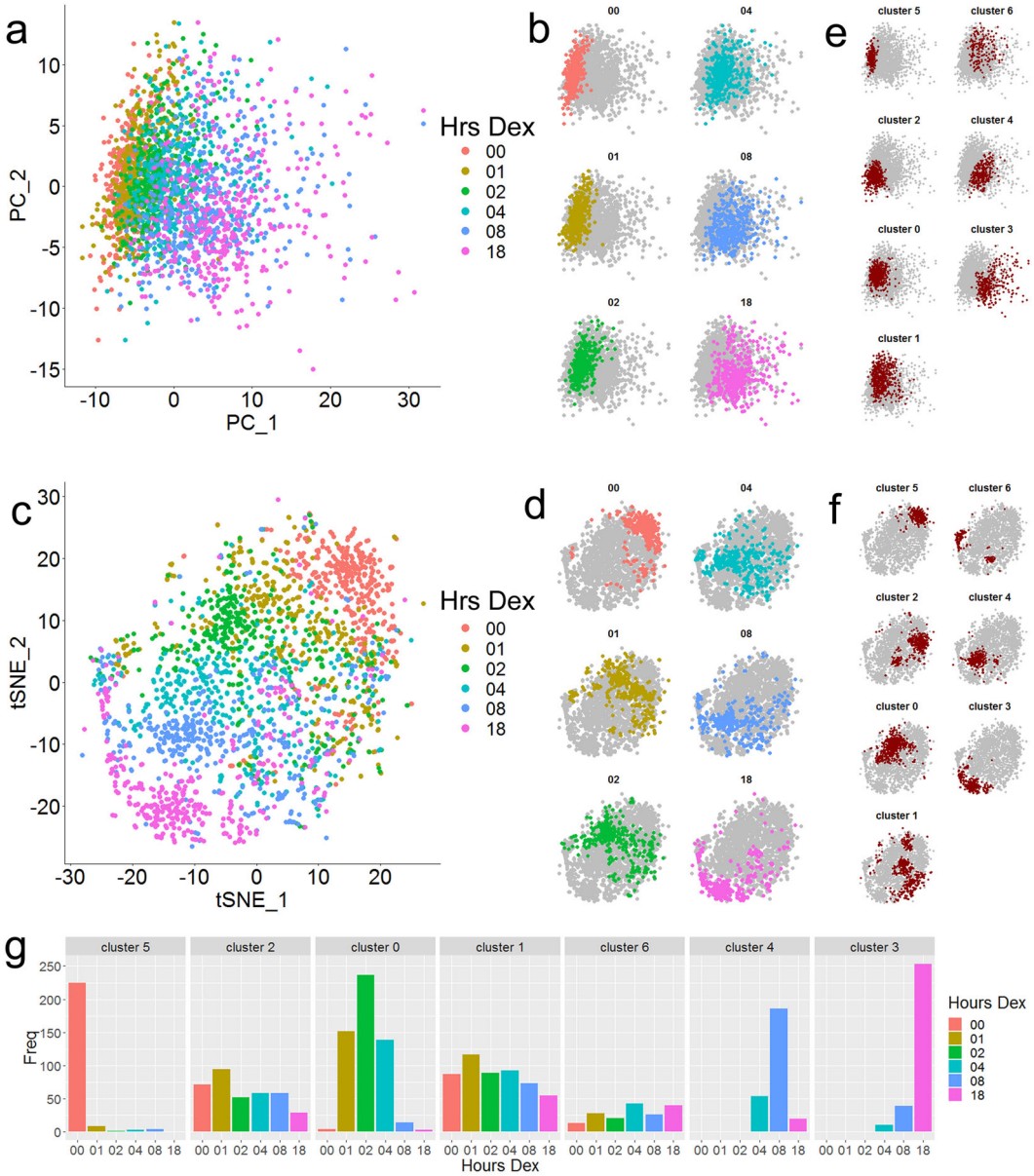

**Fig. 3 Dimensional reduction captures the transcriptional response to Dex. a** PCA plot of all 2400 cells from the scRNAseq color-coded by treatment timepoint. **b** PCA plots depicting each timepoint individually colored against all other cells. **c** tSNE plot of all 2400 cells from the scRNAseq color-coded by treatment timepoint. **d** tSNE plots depicting each timepoint individually colored against all other cells. **e**, **f** PCA and tSNE plots depicting de novo PCA-based clusters. **g** Histograms depicting the number of cells from each treatment timepoint present in the de novo PCA-based clusters.

cell exhibited a transcriptional hormone response at less than one-third of all potential Dex target genes.

As the RRG increased with prolonged Dex treatment, we reasoned that it could be driving the clustering of cells in our dimensional reductions. Mapping the RRG data as the color of the PCA plot resulted in a strong gradient from left to right, demonstrating that PC1 was driven by the RRG (Fig. 4d). Thus, the greatest source of variance within the scRNAseq dataset was the heterogeneity in the transcriptional response to Dex. Mapping the RRG as the color on the tSNE plot had similar results, with non-responsive EtOH-treated cells with the lowest RRGs clustered at the top right and the Dex-treated cells with the highest RRGs being positioned at the bottom left (Fig. 4e). Interestingly, the cells with the highest RRGs (RRG > 0.5) separated into two distinct clusters (Fig. 4e), suggesting that among the most strongly responding cells, the Dex response was

non-uniform and that there were potentially different types of Dex response.

Approaching the question of heterogeneity in the Dex-response from a gene-centric standpoint yielded similar results. Using the same SD thresholds, we determined the percentage of cells that exhibited a response for each gene. The mean responding cells percentage (RC%) at each timepoint was the same as the mean RRG (Supplemental Fig. 5). However, the range of RC% after Dex treatment was much wider than the RRG. Several genes exhibited a RC% of 0% at specific timepoints, indicative of the fact that some Dex-target genes were only detected at specific timepoints. On the other hand, *FKBP5* topped out at a RC% of 99% at 8 h (Supplemental Fig. 5) as discussed above. This demonstrated that, for each Dex-regulated gene, there was considerable variability in the proportion of Dex-treated cells that exhibited a hormone response at specific timepoints in the induction.

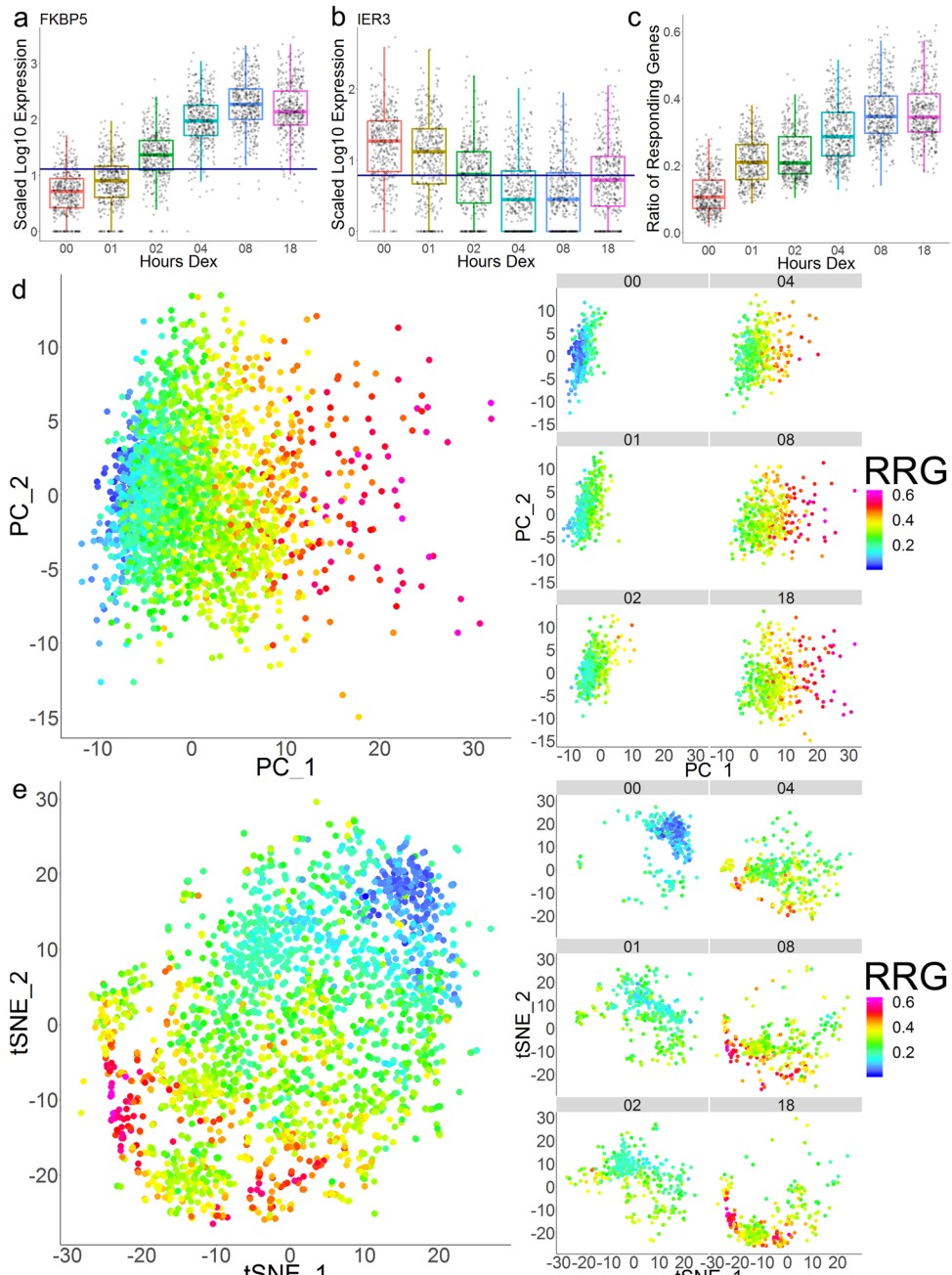

**Fig. 4 Heterogeneity in the transcriptional response to Dex. a** Box and scatter plot for FKBP5. The horizontal line indicates 1 standard deviation above the mean log-scaled expression level in EtOH-treated cells. EtOH-treated cells in which FKBP5 was not detected were excluded from this calculation. **b** Box and scatter plot for IER3. Horizontal line indicates 1 standard deviation below the mean log-scaled expression level in EtOH-treated cells. EtOH-treated cells in which IER3 was not detected were excluded from this calculation. **c** Box and scatter plot depicting the Ratio of Responding Genes for all cells in the scRNAseq. **d** PCA plots colored by the Ratio of Responding Genes for all timepoints together and each timepoint individually. **e** tSNE plots colored by the Ratio of Responding Genes for all timepoints together and each timepoint individually.

Overlaying gene expression data on tSNE plots further highlighted that the transcriptional Dex response varied between individual cells within each timepoint (Supplemental Fig. 6a). Strikingly, this variability did not appear to be driving cluster formation within the tSNE dimensional reduction (Fig. 4d), as the pattern of variability was different for each gene and highly responsive cells were frequently positioned adjacent to non-responsive cells (Supplemental Fig. 6a). Thus, there appeared to be stochastic variability in the transcriptional Dex response within the populations of cells examined at each timepoint.

To examine the relationships between Dex-regulated genes and determine whether the heterogeneity within the Dex response was entirely stochastic, we used the scaled data for each DEG to calculate Pearson correlation coefficients for every pair of DEGs across the time course of Dex treatment. Compared to a random selection of 400 genes expressed at similar levels as the scRNAseq DEGs, the range of Pearson correlations was much broader for the DEGs, with a $p$-value of 0 as calculated by Kolmogorov–Smirnov test (Supplemental Fig. 6b). The SD of correlations between DEGs was ~0.143. By comparison, the SDs

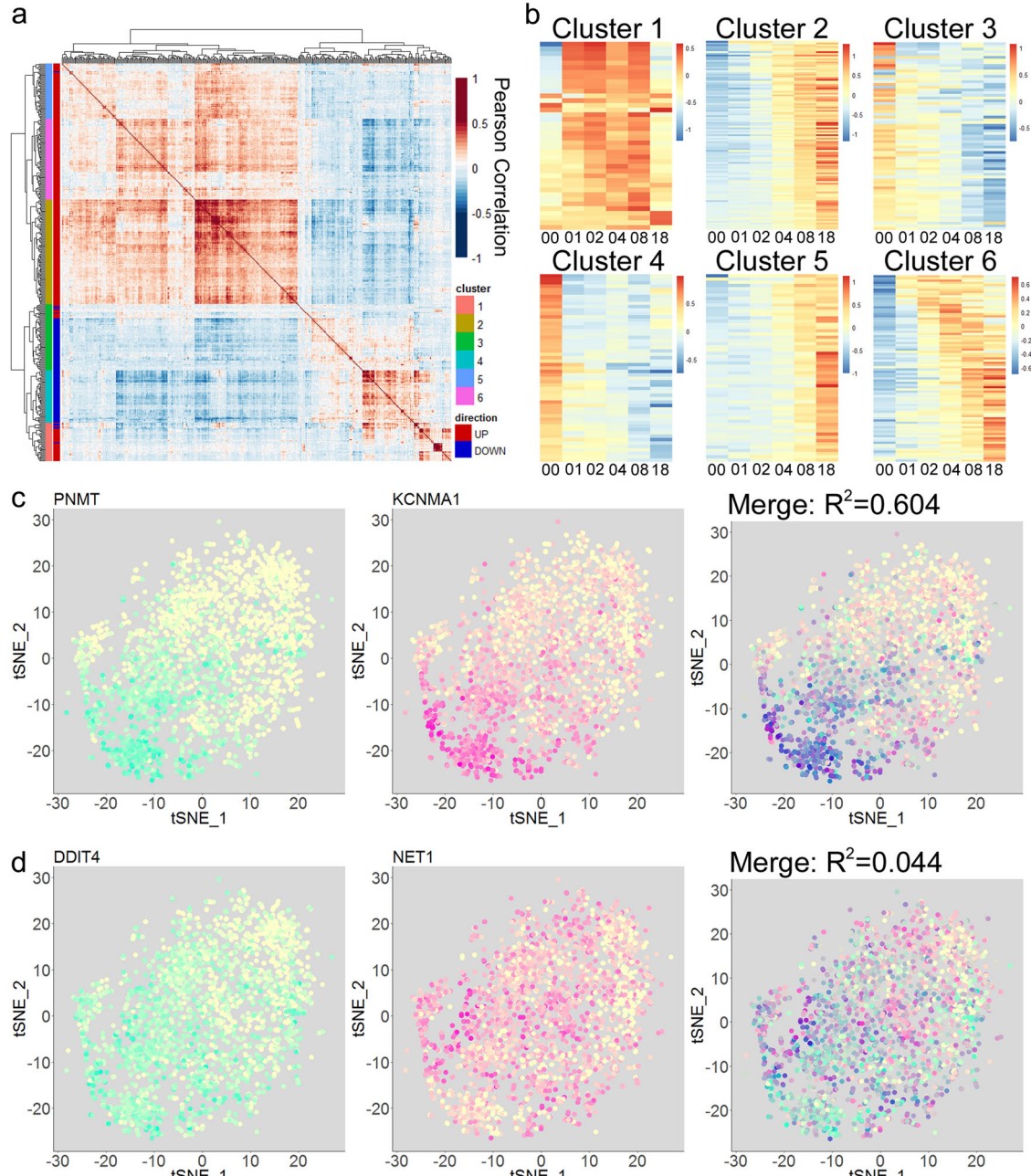

**Fig. 5 Heterogeneity and correlation in GR target gene expression patterns. a** Heatmap depicting pairwise Pearson correlations between scRNAseq DEGs. Vertical bars at the left side of the heatmap depict 6 de novo clusters derived by k-means clustering, as well as the direction in which each DEG was regulated by Dex treatment (UP = upregulated by Dex, DOWN = downregulated by Dex). **b** Heatmaps depicting the mean scaled-log expression levels in the scRNAseq for each gene cluster in **a**. **c**, **d** tSNE plots designed to mimic co-localization in immunofluorescent images. In the merged image, dark blue indicates a high level of co-expression. $R^2$ value represents the Pearson correlation of the two genes. **c** PNMT (cyan) and KCNMA1 (magenta) scaled-log expression modeled on a tSNE plots. **d** DDIT4 (cyan) and NET1 (magenta) scaled-log expression modeled on a tSNE plots.

of 1000 random groups of 400 similarly expressed genes all fell between 0.036 and 0.051 (Supplemental Fig. 6c). This indicated that expression patterns of Dex-regulated DEGs in single cells were more strongly correlated than non-regulated genes. When these pairwise correlation scores were plotted as a heatmap and hierarchical clustering was performed, the DEGs were grouped into 6 clusters (Fig. 5a). These clusters appeared to be based largely on the temporal pattern of the DEG response. Cluster 1 was largely made up of DEGs showing rapid Dex-induction at 1 h, clusters 3 and 4 contained most downregulated DEGs, and most genes in clusters 2, 5, and 6 were upregulated progressively

over the time course or at induced at later timepoints (Fig. 5b). Thus, gene-to-gene correlations appeared to be largely driven by the overall patterns of Dex-regulation rather than cell-to-cell consistency. Overlaying the expression of two genes to generate tSNE plots akin to co-immunofluorescence images revealed that even among highly correlated DEGs such as *PNMT* and *KCNMA1* ($R^2 = 0.604$), there was considerable cell-to-cell heterogeneity (Fig. 5c, additional examples of correlated genes are included in Supplemental Fig. 7). While Dex induced high levels of both genes in some cells (dark blue), there were many cells in which PNMT (cyan) or *KCNMA1* (magenta) were

predominantly induced, and many cells had intermediate levels of both genes. Conversely, with two poorly correlated DEGS like *DDIT4* and *NET1* ($R^2 = 0.044$), very few cells showed strong response at both genes (dark blue). Taken together, these analyses reveal a striking degree of cell-to-cell heterogeneity in the transcriptional response to glucocorticoid hormones.

## Discussion

Genomic experiments have demonstrated that Dex treatment alters the expression of hundreds to thousands of genes in bulk populations of cells. In this study, we utilized a single-cell transcriptomic analysis of the Dex response to show that the transcriptional response in individual cells is heterogeneous. Within a population of Dex-treated cells, each cell exhibits a unique transcriptional response. Overall, we identified ~400 Dex-regulated genes by scRNAseq across an 18 h Dex time course. However, within the average individual cell, merely 30% of these Dex-regulated genes exhibit a transcriptional response. Thus, scRNAseq revealed a striking degree of cellular heterogeneity underlying the Dex response. Furthermore, 139 Dex-regulated genes were uniquely detected by scRNAseq, providing insight into the transcriptional programs regulated by glucocorticoid signaling. These findings have wide-ranging implications for the field of hormone biology and invite several open questions about the nature of transcription overall, as well as in response to hormone treatments.

The ability to detect highly variable genes or genes that only respond in a small number of cells is a great advantage of scRNAseq in studying the transcriptional response to Dex. As each cell examined serves as a replicate, scRNAseq allows for much higher levels of statistical confidence than bulk RNAseq. This enabled for the discovery of target genes that failed detection in bulk RNAseq due to fold changes below the typically used 1.5 fold cutoff or with false-discovery rates above 0.05. This was illustrated by the identification of *ZNF703*, the oncogenic driver within the 8p12 amplification in breast cancer[17–19], as a Dex target gene in A1–2 cells. The ability to identify novel DEGs with critical functions strongly demonstrates the utility of scRNAseq in profiling the response of cancer cells to hormone stimuli or other signaling pathways. Combining the identification of previously unidentified target genes with the characterization of cellular transcriptional heterogeneity, our study reveals that, even among populations of cells thought to be homogenous, scRNAseq has considerable power to provide insight into the biology underlying cellular function and pathogenesis.

Our findings raise a critical question: what determines whether a gene will respond to hormone in a cell? One salient hypothesis is that unique response patterns in each cell are caused by heterogeneity in the patterns of GR binding to chromatin. Recent work has demonstrated that GR interacts with chromatin at very short time scales (1–10 s), and that at any given time <50% of GR molecules interact with chromatin and fewer than 10% form long-lived interactions[23]. Furthermore, individual cells appear to have, at most, tens of GR-chromatin interactions at any given time[23]. Additionally, utilization of an obligate tetrameric GR that binds to chromatin more efficiently increased the number of genes that responded to Dex treatment[24]. Thus, GR interaction with chromatin may be a limiting factor in the transcriptional response to Dex. Bulk ChIP-seq experiments suggest that GR can bind to nearly 30,000 sites in A1–2 cells[16]. However, if GR is only binding to a fraction of these sites at a time and binding site choice is largely stochastic, then transcriptional heterogeneity would be the expected output.

Another potential explanation for the cellular heterogeneity in the hormone response is that Dex-regulated genes exist in variable and stochastic transcriptionally repressive states. Live-cell single-molecule imaging of RNA produced by the Estrogen-responsive gene *TFF1* in MCF7 cells revealed that in response to estradiol, RNA synthesis is infrequent and irregular[14]. Similar to what we observe here, individual cells were shown to exhibit a wide range of *TFF1* induction, with some cells rapidly inducing high levels, and some cells not inducing *TFF1* at all[14]. The variability in TFF1 RNA synthesis was attributed to the existence of persistent transcriptionally repressive states, suggesting that transcriptional heterogeneity was driven by underlying stochastic heterogeneity in the epigenetic landscape surrounding gene promoters or other regulatory elements.

Precedent for repressive chromatin states in hormone signaling exists in work demonstrating that prolonged Dex exposure results in a chromatin-dependent refractory phase where Dex-responsiveness is lost at the MMTV promoter[25]. This appeared to be caused by a loss of histone H1 phosphorylation, suggesting that modified chromatin environments around GR-binding sites blocked hormone responsiveness[26]. Furthermore, brief hormone treatments can have long-lasting effects on the response characteristics of cells. Indeed, chromatin dynamics induced by glucocorticoid treatment persist for variable lifetimes following hormone withdrawal, with some alterations persisting for days[27,28]. In HepG2 human liver carcinoma cells, ApoB and ApoE are induced by 24 to 48 h Estrogen treatments. However, after removal of Estrogen, response kinetics were altered for up to 15 generations/cell divisions after the original Estrogen treatment[29]. Thus, dynamic and potentially long-lasting chromatin states do appear to regulate the transcriptional response to hormone.

Recent advances in the development of single-cell epigenomic assays have begun to provide many of the tools needed to address these hypotheses. For instance, single-cell ChIP-seq (scChIPseq) and single-cell ATAC-seq (scATACseq) allow for the examination of the patterns of active and repressive chromatin landscapes in individual cells[30,31]. These studies have revealed that chromatin states are indeed heterogeneous across populations of cells. Profiling the chromatin landscape across Dex-treated cells could address whether heterogeneity in the underlying chromatin landscape or Dex-induced chromatin remodeling contribute to the heterogeneity in the transcriptional response to Dex. Beyond profiling the chromatin landscape, CUT&Run and CUT&Tag protocols are capable of characterizing transcription factor binding in single cells[32,33]. Examining GR interaction with chromatin and how the chromatin landscape changes in response to Dex in single cells with these methods could definitively address the mechanisms that give rise to this heterogeneity.

## Methods

**Cell culture**. T47D A1–2 cells were cultured as previously described[34], authenticated by STR profiling, and tested negative for mycoplasma. For Dexamethasone treatments, cells were initially cultured for 24 h in reduced-serum, hormone-stripped media (Phenol-red free MEM (Gibco 51200) with 5% Charcoal/Dextran treated FBS (Atlanta S11650), 1X Penicillin/Streptomycin (Sigma P0781), 1% HEPES (Sigma H0887), 1X Glutamax (Gibco 35050), and 250 ug/ml G418 (Gibco 10131)). Subsequently, Dexamethasone (Sigma D4902) was added to the media at 100 nM for all treatments timepoints and $9.5 \times 10^{-4}$% ethanol was used as vehicle control. For both RNAseq experiments, control cells were treated with ethanol for 18 h.

**RNA isolation, cDNA synthesis, QPCR, and RNAseq.** Qiagen RNeasy kits with on-column DNase treatment were used per manufacturer's instructions to isolate total RNA from A1–2 cells in biological triplicate. Thermofisher SuperScript III was used following manufacturer's instructions with Oligo (dT) to synthesize complementary DNA (cDNA) from 1 ng total RNA. BioRad ssoAdvanced Universal SYBR Green Supermix was used for QPCR. Prior to bulk RNAseq, RNA quality was confirmed on the Agilent Bioanalyzer 2100 using the RNA 6000 RNA Pico Kit. Total RNAseq libraries were generated in the NIEHS Epigenomics Core using Ribo-Zero Gold and sequenced on the Illumina NovaSeq 6000 to over 140 million

reads per biological replicate. Bulk RNAseq data was processed as described previously[16].

**Single-cell RNAseq.** To create single-cell suspensions for scRNAseq, cells were dissociated using 0.25% Trypsin-EDTA (Gibco 25200). The BioRad ddSeq Single-Cell Isolator was used to create single-cell emulsions and the Illumina SureCell WTA 3′ Library Prep kit was used to process samples and generate scRNAseq libraries. In order to detect as many genes as possible, libraries were sequenced at high depth on the Illumina Novaseq 6000 to obtain >200 million raw reads for each library. Reads were parsed for cell barcode and UMI information, filtering out reads according to the following constraints: no. 6 nucleotide barcode segment is more than a Hamming distance of 1 from a valid barcode block; all linker segments are within 1 nucleotide length of the expected value; the UMI has a length of 8 nucleotide; and only 1 mismatch is allowed within the ACG-GAC segments flanking the UMI. Reads with cell barcode information passing these filters were then aligned to the hg19 reference genome using STAR v2.5.2b[35] and annotated with the Gencode comprehensive v28lift37. Uniquely mapped reads were subsequently assigned to genes using featureCounts v1.5.3[36]. PCR duplicates were removed using umi_tools[37] in per-gene mode. The knee-calling algorithm from umi_tools was used to identify an appropriate total transcript count cutoff for cells in each sample, yielding a total of 4001 cells. Cells with mitochondrial gene percentage >5% were removed, and cell counts were balanced between timepoints by randomly sub-setting each timepoint to 400 cells. Cyclone from the scran package[38,39] was used to determine cell-cycle scores for all cells, and Seurat v3[40] was then use to normalize and scale the data such that the impacts of transcript counts, mitochondrial percent, and cell-cycle scores were regressed-out. Seurat and ggplot2[41] were used for data visualization.

**Statistics and reproducibility.** Bulk RNAseq was performed using independent biological triplicates of A1–2 cells at each timepoint of Dex treatment. For scRNAseq, data analysis was performed using a randomly down-sampled set of 400 A1–2 cells for each timepoint. Statistical analysis was performed in R using Seurat for scRNAseq and using Limma-Voom for bulk RNAseq. Three biological replicates were used for RT-QPCRs and ChIP-QPCRs depicted in Fig. 2f, h, and p-values were calculated in Excel using one-tailed, heteroscedastic t-tests. Pearson correlations of gene expression depicted in Fig. 5 were calculated in R.

**Reporting summary.** Further information on research design is available in the Nature Research Reporting Summary linked to this article.

## Data availability
RNAseq data generated for this publication have been deposited in NCBI's Gene Expression Omnibus[42] and are accessible through GEO Series accession number GSE141834. Our previously published dataset is accessible through GSE112491.

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

## Acknowledgements

We thank members of the Archer Group and the NIEHS Integrative Bioinformatics Group for critical review of the data and manuscript and for their ongoing support. We also thank Xin Xu, Molly Cook, Nicole Reeves, Jason Malphurs, and Greg Solomon of the NIEHS Epigenomics Core Laboratory for library preparation and next generation sequencing expertize. Finally, we thank David Fargo, Humphrey Yao, and Joseph Rodriguez for critical review of the manuscript.

## Author contributions

J.A.H. and T.K.A. conceptualized and designed the project. J.A.H. and K.W.T performed experiments. J.A.H. and B.N.P. performed bioinformatic data analysis. J.A.H. made figures and wrote the manuscript. B.N.P., K.W.T., and T.K.A. reviewed the manuscript. T.K.A. acquired funding and supervised the project. This research was supported by the Intramural Research Program of the NIH, National Institute of Environmental Health Sciences (Z01 ES071006-20).

## Competing interests

The authors declare no competing interests.
