## [Peer Review File · Communications Biology]

Reviewers' comments:

Reviewer #1 (Remarks to the Author):

In this manuscript, the authors present the results of a single cell RNA sequencing (scRNAseq) analysis of T47D A1-2 human breast cancer cells at different time points after treatment with dexamethasone (dex), a specific glucocorticoid receptor (GR) agonist. The authors have compared the results of the scRNAseq (400 cells per time point) with the results of a similar experiment in which bulk RNAseq was performed. Comparing bulkRNAseq and scRNAseq, the latter showed a bias towards identification of transcripts from highly expressed genes, and fewer differentially expressed genes (DEGs) were identified. However, about one third of the DEGs found in scRNAseq were not detected by bulk RNAseq (mainly genes with a milder or more variable dex effect), indicating that this approach is able to reveal an additional subset of dex-responsive genes. Comparison with Chromatin Immunoprecipitation sequencing (ChIPseq) data showed that GR binding close to bulkRNAseq and scRNAseq-specific genes was similar. Principal component analysis (PCA) demonstrated that the dex response was well represented on the first principal component, having cells of the same time point clustering together, although still some heterogeneity remained in the response. At 8 and 18 hours of treatment, on average ~35% of all genes were regulated, with considerable variation. Furthermore, for each dex-regulated gene large variation was found in the size of the population of cells that showed a significant response. Finally, by correlating the expression levels of individual genes, clusters of correlated genes were identified, which appeared to be largely driven by a common time-dependent pattern of dex regulation.

Single-cell RNA sequencing is a relatively novel technology and the authors nicely demonstrate in this manuscript the added value as well as the limitations of the approach. In addition, the data shed new light on the heterogeneity of the transcriptional response to dex treatment. The results are thoroughly and carefully analyzed and the manuscript is very well and clearly written. However, I have one major concern, which could be addressed by rewriting parts of the Results section. In addition, I would like to raise several minor issues, mostly out of curiosity rather than criticism.

Major issue:

Starting at line 249, the authors discuss the 'Cellular variability within the hormone response', but they actually do not determine the response of individual cells to dexamethasone because they do not know the gene expression levels of the individual cells before the dex treatment. Instead, the authors calculate the ratio between the expression level of an individual cells and the average expression level in the control group. This should not be called the individual response to dex, but it represents the relative expression level of an individual cell. It is conceivable that low expression levels of a gene 18 hours after dex treatment result from a high response since the initial expression could have been low. I think the authors should carefully rewrite the Results section with this in mind.

Minor issues:

- Fewer genes are identified in scRNAseq, but how does this relate to the depth of sequencing? More genes will be identified in scRNAseq if the depth of sequencing will be increased. Please discuss.
- The comparison of DEGs between scRNAseq and bulkRNAseq (lines 112-124) makes no sense to me when different cutoffs for p value and fold change are used. Since the number of DEG is largely dependent on these cutoffs, comparing the number of DEGs is meaningless (although it is still interesting that one third of DEGs in scRNAseq is not found in bulk RNAseq). For proper comparisons, one could compare volcano plots p value against fold change, or make scatter plots showing the relation between p values (or fold changes) in scRNAseq and bulkRNAseq.
- Line 183: 'more' instead of 'greater'.
- The control group was treated for 18 hours with Vehicle (Ethanol). It could be argued that this is not

a proper control for the other time points (although taking a vehicle control for each time point would be asking too much). At least the authors should refer to this group as 'control' in all their figures (so not as '0', which could be considered misleading).

- How does technical reproducibility add to the observed heterogeneity? How would the heterogeneity look if 400 times the same cell would be sequenced?
- What do we know about heterogeneity in tissues in a living organism? How do the authors think their results relate to an in vivo situation? Please discuss.
- In the Discussion the authors offer two possible explanations for the observed heterogeneity. The first possible explanation is the stochastic nature of DNA binding by GR due to short binding times. However, at the time scales of hours the differences in binding should be averaged out, because of the very large number of individual (short) binding events.
- The second possible explanation is the stochastic nature epigenetic changes in the chromatin landscape. However, the correlations between different genes shown in Figure 5 shows that this is not entirely stochastic and certain patterns exist.
- Is it possible to state that cell-to-cell variation is the largest contributor to the observed heterogeneity and that gene-to-gene variation plays a smaller role?

Reviewer #2 (Remarks to the Author):

Does the manuscript have technical or conceptual flaws that should prohibit its publication? If so, please provide details.

The manuscript does not have technical or conceptual flaws that prohibit its publication.

Are the conclusions original? Somewhat

If not, please provide relevant references. Additional references are provided below.

Do you feel that the results presented are of immediate relevance for many people in your own discipline or for a broader audience? Mostly relevant to people in the field and somewhat to the broader audience.

If you recommend publication, please outline briefly what you consider to be the outstanding features. If could be published after the modifications suggested below.

If you feel that specific additional experiments would strengthen the case for publication in Communications Biology, please provide suggestions. Suggestions for an additional experiment and more analysis are provided below.

In the manuscript "Transcriptional Heterogeneity within the Glucocorticoid Response" Hoffman et al., examine glucocorticoid receptor (GR)-mediated response in individual cells over different activation times (0, 1, 2, 4, 8 and 18h) by single cell RNA sequencing (scRNAseq). By comparing the response of bulk populations of cells to the response in individual cells they were able to identify over 100 GRregulated genes that were not detected in bulk RNA-seq. They also demonstrate that on average only

30% of the total set of GR regulated genes are detected simultaneously at a single cell level. The authors

concluded that transcriptional response to hormone is highly variable in individual cells.

Experiments are conducted and analyzed properly and provide additional evidence for the previously demonstrated transcriptional heterogeneity upon steroid activation. These findings are of interest for the people in the field of steroid hormone signaling and might also be of interest to a broader audience.

Major points:

A version of the scRNAseq, namely single cell combinatorial indexing RNA sequencing (sci-RNA-seq) was

already used to demonstrate transcriptional heterogeneity at a single cell level after 0, 1, and 3h of glucocorticoid stimulation in a cell line (A549), as well as in individual cells from adult mouse kidney (see

Cao et al., Science, 2018: 1380-1385). The study is not cited by the manuscript under review, which is

major omission, and should be included as a prior work on the subject.

Citation of papers demonstrating cellular heterogeneity in response to glucocorticoid treatment is also somewhat incomplete and we recommend inclusion of a recent work demonstrating heterogeneous activation of the MMTV-driven pp7 stem-loops reporter in living cells (Stavreva et al., Mol. Cell 2019). The possibility that ZNF703 gene upregulation by glucocorticoid treatment has important implications in

breast cancer etiology but this hypothesis was not evaluated further. This reviewer would like to see a follow-up experiment testing the effects of the Dex treatment (and presumably ZNF703 gene activation)

on ER-mediated transcriptional responses.

The authors examine in detail the data for GR-induced genes while largely ignoring the examples for the

GR-repressed genes. Considering the substantial repressive role of GR in anti-inflammatory processes these analyses will be of major interest and should be included.

The finding that "Dex-regulated DEGs in single cells were more strongly correlated than non-regulated genes" (example in Figure 5C) is interesting and novel and should be elaborated by providing further examples. It should also be highlighted in the Abstract and the Discussion.

In the Discussion section (first paragraph on page 18) authors discuss the long-lasting effects of a brief

hormonal treatments on chromatin using the estrogen receptor activation as an example. Locus-specific,

long-lasting changes in chromatin structure after short glucocorticoid treatments were also demonstrated (Stavreva et al., Genome Res. 2015 and Jubb et al., Cell Rep. 2017) and these studies should be cited in the manuscript.

Figure 1E: It is unclear how many samples of bulk RNA-seq was used for the analyses. Error bars and statistical analysis should be included.

Figure 2F: The RT-qPCR data validating the Dex-dependent transcriptional upregulation of the ZNF703 gene are missing the statistical analysis for the significance of the observed changes.

Figure 2H: As above, statistical analysis for the significance of the observed changes are missing.

Minor points:

Methods do not specify for how long the cells were pre-treated with hormone-stripped media. This is critical for the interpretation of the data as a short pre-treatment will lead to a higher RNA-seq read counts for GR-regulated genes in the control.

The quality of the right panels (the individual timepoint PCA plots) in Figure 4 as well as the panel in Figure 5A is poor. Those panels should be replaced with better-quality pictures.

Reviewers' comments:

Reviewer #1 (Remarks to the Author):

In this manuscript, the authors present the results of a single cell RNA sequencing (scRNAseq) analysis of T47D A1-2 human breast cancer cells at different time points after treatment with dexamethasone (dex), a specific glucocorticoid receptor (GR) agonist. The authors have compared the results of the scRNAseq (400 cells per time point) with the results of a similar experiment in which bulk RNAseq was performed. Comparing bulkRNAseq and scRNAseq, the latter showed a bias towards identification of transcripts from highly expressed genes, and fewer differentially expressed genes (DEGs) were identified. However, about one third of the DEGs found in scRNAseq were not detected by bulk RNAseq (mainly genes with a milder or more variable dex effect), indicating that this approach is able to reveal an additional subset of dex-responsive genes. Comparison with Chromatin Immunoprecipitation sequencing (ChIPseq) data showed that GR binding close to bulkRNAseq and scRNAseq-specific genes was similar. Principal component analysis (PCA) demonstrated that the dex response was well represented on the first principal component, having cells of the same time point clustering together, although still some heterogeneity remained in the response. At 8 and 18 hours of treatment, on average ~35% of all genes were regulated, with considerable variation. Furthermore, for each dex-regulated gene large variation was found in the size of the population of cells that showed a significant response. Finally, by correlating the expression levels of individual genes, clusters of correlated genes were identified, which appeared to be largely driven by a common time-dependent pattern of dex regulation.

Single-cell RNA sequencing is a relatively novel technology and the authors nicely demonstrate in this manuscript the added value as well as the limitations of the approach. In addition, the data shed new light on the heterogeneity of the transcriptional response to dex treatment. The results are thoroughly and carefully analyzed and the manuscript is very well and clearly written. However, I have one major concern, which could be addressed by rewriting parts of the Results section. In addition, I would like to raise several minor issues, mostly out of curiosity rather than criticism.

Major issue:

Starting at line 283, the authors discuss the 'Cellular variability within the hormone response', but they actually do not determine the response of individual cells to dexamethasone because they do not know the gene expression levels of the individual cells before the dex treatment. Instead, the authors calculate the ratio between the expression level of an individual cells and the average expression level in the control group. This should not be called the individual response to dex, but it represents the relative expression level of an individual cell. It is conceivable that low expression levels of a gene 18 hours after dex treatment result from a high response since the initial expression could have been low. I think the authors should carefully rewrite the Results section with this in mind.

We appreciate the reviewer's concern. We have taken care to re-write/add to the beginning of this section of the results (lines 289-292) to make it clear that, while we are not directly measuring the response (because we cannot measure the expression levels at two timepoints in single cells), we are determining whether Dex-treated cells have express Dex target genes at levels outside of the normal range of expression in untreated cells.

Minor issues:

- Fewer genes are identified in scRNAseq, but how does this relate to the depth of sequencing? More genes will be identified in scRNAseq if the depth of sequencing will be increased. Please discuss.

The reviewer raises an excellent point that concerned us greatly when initiating this project. In collaboration with other labs in our department, we conducted a pilot experiment to “test the limits” of our scRNAseq platform. We iteratively sequenced scRNAseq libraries to determine whether more sequencing would yield more genes. Initially this was the case, but beyond 200 million reads, identification of new UMIs and genes was greatly reduced and not at all cost-effective. For the dataset in this manuscript, we obtained over 200 million reads for each timepoint, so we are confident that our sequencing depth is a close to saturation as economically possible. We have edited a sentence in our methods section (lines 546-548) to indicate this:

“In order to detect as many genes as possible, libraries were sequenced at high depth on the Illumina Novaseq 6000 to obtain more than 200 million raw reads per library.”

- The comparison of DEGs between scRNAseq and bulkRNAseq (lines 126-138) makes no sense to me when different cutoffs for p value and fold change are used. Since the number of DEG is largely dependent on these cutoffs, comparing the number of DEGs is meaningless (although it is still interesting that one third of DEGs in scRNAseq is not found in bulk RNAseq). For proper comparisons, one could compare volcano plots p value against fold change or make scatter plots showing the relation between p values (or fold changes) in scRNAseq and bulkRNAseq.

The reviewer raises an important point and we certainly appreciate the concern. As the two datasets have remarkably different sample numbers (3 per treatment for bulk RNAseq, and essentially 400 per treatment for scRNAseq), it is difficult to compare the stringency of significance cutoffs. We experimented with many different Pvalue and fold change cutoffs with both data sets, including using the same cutoffs for both. Across the board, the approximate ratio of DEGs in the bulk vs the scRNAseq was similar. However, altering the cutoffs would result in either false positives based on an “eye test” of the DEGs just clearing the significance cutoffs or false negatives of known DEGs that lost statistical significance with increased stringency. As such, we chose to proceed with the “default” fold change and Pvalue cutoffs for each data set based on the recommendations of the software used for data analysis and a wealth of previously published research.

- Line 208: ‘more’ instead of ‘greater’.

This change has been made in the revised manuscript

- The control group was treated for 18 hours with Vehicle (Ethanol). It could be argued that this is not a proper control for the other time points (although taking a vehicle control for each time point would be asking too much). At least the authors should refer to this group as ‘control’ in all their figures (so not as ‘0’, which could be considered misleading).

We appreciate the issue that the reviewer’s raises here. In previous experiments, we have performed EtOH controls for each timepoint, and in all circumstances, EtOH timepoints were indistinguishable from each other. Thus, we feel confident that our EtOH control group represents the baseline of gene expression for all timepoints of Dex treatment. However, to avoid any misunderstanding, we have added text to the beginning of the Results section (lines 73-75) to clarify that all timepoints were collected at the same time and that we are referring to the “control” cells as “0” in our figure because they receive no Dex treatment.

- How does technical reproducibility add to the observed heterogeneity? How would the heterogeneity look if 400 times the same cell would be sequenced?

This is a very interesting question. Hypothetically, we would guess that sequencing the same cell 400 times would yield much more homogeneous results. While genes detected at very low levels would be subject to technical variability between replicates of the same cell, we would expect that robustly expressed genes would be consistently detected across technical replicates of the same cell.

- What do we know about heterogeneity in tissues in a living organism? How do the authors think their results relate to an in vivo situation? Please discuss.

There is definitely expression heterogeneity in living tissues. One such example is discussed/examined in the Rodriguez et al. 2019, Cell paper that we reference in our manuscript (Ref. 14). We would expect that an examination of heterogeneity in the Dex response in vivo would yield similar results to the ones that we report here. We are hopeful that anticipated studies with mouse tissue samples will provide even greater insight into the source of transcriptional heterogeneity, demonstrated in our single cell experiments.

- In the Discussion the authors offer two possible explanations for the observed heterogeneity. The first possible explanation is the stochastic nature of DNA binding by GR due to short binding times. However, at the time scales of hours the differences in binding should be averaged out, because of the very large number of individual (short) binding events.

It is certainly possible that differences in GR binding could be averaged out over the course of hours of Dex treatment, but as of yet, no one has directly tested this hypothesis. We hope to do so in the future with by performing some of the single-cell epigenomic assays that we describe in the final paragraph of the discussion section. Until then, we can neither confirm nor rule out this explanation for the transcriptional heterogeneity in the Dex response.

- The second possible explanation is the stochastic nature epigenetic changes in the chromatin landscape. However, the correlations between different genes shown in Figure 5 shows that this is not entirely stochastic and certain patterns exist.

While patterns and correlations do exist in the expression of Dex-regulated genes, these are not absolute and there remains a great deal of heterogeneity and apparent stochasticity in the transcriptional response. In fact, the correlations appear to be driven more by the temporal pattern of each gene's expression than the cell-to-cell correlation of expression levels. As we reference in the manuscript, there is evidence that cell-to-cell variability in the chromatin landscape could be responsible for transcriptional heterogeneity. As such and as above, we hope to follow up on this work by designing experiments to determine whether this explanation is valid.

- Is it possible to state that cell-to-cell variation is the largest contributor to the observed heterogeneity and that gene-to-gene variation plays a smaller role?

We think that this is probably the case. Whether either, both, or neither of the two potential explanations for heterogeneity that we proposed are proven to be valid, it does seem that cell-intrinsic differences are the cause. However, different genes may be more or less prone to being affected by cell-

to-cell variability. Again, this is a very interesting question, and it is one that we (and certainly many other labs) will be attempting to address in the future.

Reviewer #2 (Remarks to the Author):

Does the manuscript have technical or conceptual flaws that should prohibit its publication? If so, please provide details. The manuscript does not have technical or conceptual flaws that prohibit its publication.

Are the conclusions original? Somewhat

If not, please provide relevant references. Additional references are provided below.

Do you feel that the results presented are of immediate relevance for many people in your own discipline or for a broader audience? Mostly relevant to people in the field and somewhat to the broader audience.

If you recommend publication, please outline briefly what you consider to be the outstanding features. It could be published after the modifications suggested below.

If you feel that specific additional experiments would strengthen the case for publication in *Communications Biology*, please provide suggestions. Suggestions for an additional experiment and more analysis are provided below.

In the manuscript "Transcriptional Heterogeneity within the Glucocorticoid Response" Hoffman et al., examine glucocorticoid receptor (GR)-mediated response in individual cells over different activation times (0, 1, 2, 4, 8 and 18h) by single cell RNA sequencing (scRNAseq). By comparing the response of bulk populations of cells to the response in individual cells they were able to identify over 100 GR regulated genes that were not detected in bulk RNA-seq. They also demonstrate that on average only 30% of the total set of GR regulated genes are detected simultaneously at a single cell level. The authors concluded that transcriptional response to hormone is highly variable in individual cells.

Experiments are conducted and analyzed properly and provide additional evidence for the previously demonstrated transcriptional heterogeneity upon steroid activation. These findings are of interest for the people in the field of steroid hormone signaling and might also be of interest to a broader audience.

Major points:

A version of the scRNAseq, namely single cell combinatorial indexing RNA sequencing (sci-RNA-seq) was already used to demonstrate transcriptional heterogeneity at a single cell level after 0, 1, and 3h of glucocorticoid stimulation in a cell line (A549), as well as in individual cells from adult mouse kidney (see Cao et al., *Science*, 2018: 1380-1385). The study is not cited by the manuscript under review, which is a major omission, and should be included as a prior work on the subject.

We thank the reviewer for pointing out this omission. We have added a reference to this paper in our introduction section (lines 54-55).

Citation of papers demonstrating cellular heterogeneity in response to glucocorticoid treatment is also somewhat incomplete and we recommend inclusion of a recent work demonstrating heterogeneous activation of the MMTV-driven pp7 stem-loops reporter in living cells (Stavreva et al., Mol. Cell 2019). This paper became available a few weeks after we submitted our manuscript. It provides an excellent example of cellular heterogeneity in response to glucocorticoids, and we have added a reference to it in our introduction (lines 48-50).

The possibility that ZNF703 gene upregulation by glucocorticoid treatment has important implications in breast cancer etiology but this hypothesis was not evaluated further. This reviewer would like to see a follow-up experiment testing the effects of the Dex treatment (and presumably ZNF703 gene activation) on ER-mediated transcriptional responses.

We are also greatly interested in the implications that glucocorticoid regulation of ZNF703 has for breast cancer etiology and in potential interactions between different hormone-mediated transcriptional responses in ZNF703 regulation. As such, we have performed short-term co-treatments with Dex and Estrogen in A1-2 cells and examine the effects on ZNF703 expression (see attached graph below). Dex alone results in a 2.5-fold upregulation of ZNF703 at 2 hours, and this induction diminishes to 1.5-fold by 8 hours. Conversely, Estrogen induces a 1.5-fold upregulation at 2 and 4 hours and increases to nearly 2-fold at 8 hours. Combined treatment resulted in an induction pattern similar to that observed in Dex alone.

We also performed staggered treatments, namely cells were treated with Dex for 8 hours and Estrogen was added for the last 4 hours (or vice versa). In both of these treatments, the induction of ZNF703 was most similar to that caused by Dex alone.

Based on these experiments, Dex seems to have the dominant effect on ZNF703 induction in this context. In future experiments, we plan to perform additional co-treatments with Dex, Estrogen, and other relevant hormone/nuclear receptor agonists to further detail how hormone signaling regulates the transcription of ZNF703. Furthermore, we will perform also perform these experiments in additional breast cancer cell lines that may be more clinically and/or genetically relevant to the types of breast cancer in which ZNF703 putatively acts as an oncogene. However, we feel that these experiments are beyond the scope of this manuscript.

The authors examine in detail the data for GR-induced genes while largely ignoring the examples for the GR-repressed genes. Considering the substantial repressive role of GR in anti-inflammatory processes these analyses will be of major interest and should be included.

We apologize if it was unclear in our description of the data analyses that we did include repressed genes in all of the analyses performed in this manuscript. However, we tried to point out in our description of the calculation of the Ratio of Responding Genes (RRG) that we were considering both GR-induced and GR-repressed genes (Results, lines 307-310):

“We called a gene “responsive” in each cell if it was expressed at a level greater than one SD above the mean EtOH-treated level **(or more than one SD below the mean for down-regulated genes)**. We performed this calculation for **all scRNAseq DEGs** to determine the RRG in each cell.”

However, we did not provide any examples of GR-repressed genes in our figures. To remedy this, we have

- 1) added a new supplemental figure 1 analogous to Figure 1E-H to demonstrate the detection of GR-repressed genes (lines 119-121 and 168-171),
- 2) added a panel to figure 4 to demonstrate the response of the repressed gene IER3 (panel 4B, lines 296-300), and
- 3) included an example of strongly correlated GR-repressed genes to a new supplemental figure 7. (Supplemental figure 7A)

The finding that “Dex-regulated DEGs in single cells were more strongly correlated than non-regulated genes” (example in Figure 5C) is interesting and novel and should be elaborated by providing further examples. It should also be highlighted in the Abstract and the Discussion.

We thank the reviewer for this suggestion. We have added a new supplemental figure 7 that includes 4 additional examples of strongly correlated gene pairs (lines 394-395 and 430-436).

In the Discussion section (first paragraph on page 18) authors discuss the long-lasting effects of a brief hormonal treatments on chromatin using the estrogen receptor activation as an example. Locus-specific, long-lasting changes in chromatin structure after short glucocorticoid treatments were also demonstrated (Stavreva et al., Genome Res. 2015 and Jubb et al., Cell Rep. 2017) and these studies should be cited in the manuscript.

Thank you for pointing out these unfortunate omissions. We have added references to these papers to the relevant paragraph of the discussion section (lines 500-501)

Figure 1E: It is unclear how many samples of bulk RNA-seq was used for the analyses. Error bars and statistical analysis should be included.

The data in Figure 1E represents the log₁₀ mean read counts of bulk RNAseq biological triplicates. The figure has been edited to include errors bars indicating the standard deviation between biological triplicates and with asterisks annotating data points that are significantly different than the control cells (p-value <0.05). The figure legend has also been edited to account for this addition.

Figure 2F: The RT-qPCR data validating the Dex-dependent transcriptional upregulation of the ZNF703 gene are missing the statistical analysis for the significance of the observed changes.

Figure 2H: As above, statistical analysis for the significance of the observed changes are missing.

We have performed the necessary statistical analysis and have updated these figures to include asterisks annotating significant data points. The figure legend has also been edited to account for these additions.

Minor points: Methods do not specify for how long the cells were pre-treated with hormone-stripped media. This is critical for the interpretation of the data as a short pre-treatment will lead to a higher RNA-seq read counts for GR-regulated genes in the control.

Thank you for pointing out this omission. For all experiments, cells were pre-treated with hormone-stripped media for 24 hours. We have edited the relevant methods passage to include this information (lines 522-529)

The quality of the right panels (the individual timepoint PCA plots) in Figure 4 as well as the panel in Figure 5A is poor. Those panels should be replaced with better-quality pictures.

Thank you for pointing this out – this is an unfortunate side effect of compressing the figures for inclusion in a single, small .pdf for initial submission. All figures will be updated with higher resolution, publication-quality figures if publication is approved.

Reviewers' comments:

Reviewer #2 (Remarks to the Author):

All my major concerns have been appropriately addressed. This version of the manuscript is of an improved quality and is suitable for publication in Communications Biology.

Reviewer #3 (Remarks to the Author):

Hoffman et al. study and describe the cellular heterogeneity in response to glucocorticoids in a breast cancer cell line, using a time course of Dex treatment. In the analysis they compare results from bulk and single cell RNA-seq data. In general I found some of the methodologies used not very conventional, for the analysis of single cell data. I am missing some specifications on the methodology and data processing, and also the reviewers token to the GEO accession. The authors show that there is heterogeneity but they do not show an unsupervised clustering of the data, which if there are significant transcriptional heterogeneous groups, should be reflected easily. In order to consider the manuscript, I would ask the authors to explain the doubts raised and to answer my concerns.

1) Is usual to find differences in the detection of DEGs when comparing bulk and single cell. Lowly expressed genes will be more difficult to be identified in single cell. Also bulk RNA-seq losses resolution of the single cell because is basically an average of the expression for all the pooled cells. The authors use a method of ranking gene expressions by quantiles to be able to compare both single and RNA-seq.

When comparing bulk and single cells DEGs, authors mention that there is a high variability between replicates in the bulk RNA-seq. Authors should explain why there is such a variability, are these technical or biological replicates. For any bulk RNA-seq experiment I would expect some correlation or PCA between the replicates. High variability could mean that these can not be used as actual replicates. The model used for differential gene expression should consider a normalization of the read counts in the cases that there is a high variability of the replicates. Can the authors explain there is such variability?

2) Authors identify ZNF703 is regulated by GR by identifying GR peaks in a region of 99Kb upstream its promoter. Then, authors verify a possible enhancer in this peak region when they investigate public data including H3K27ac. When investigating the upstream region, the authors choose a large window. If we include gencode annotations to this region, there are several lncRNAs that match exactly the position of these GR peaks. Could be that this enhancer or promoter belong to these upstream lncRNAs? Authors should compare to available Hi-C data in order to identify if there is evidence of a link between this enhancer and the promoter of ZNF703. I would like to see if there is a link between the enhancers and the promoter. This would give more strong evidence of the regulation of ZNF703 by GR. Can the authors include in the supplementary figure 3 the tracks from gencode to see any other overlapping annotated gene on the selected region?

The authors generated many sequencing data, it will be useful for the reviewers to have a genome browser link with the tracks to see the actual location of the peaks, and in this case investigate the ZNF703 region.

3) When authors mention that ZNF703 was detected in more than 95% of the cells. They are counting all the cells with expression more than 0 for this gene? Or there is any specific threshold.

4) The authors are going to deposit the data in GEO repository. It is essential that data is available for the reviewers before any decision. Therefore, I would like to have access to the reviewers token.

5) It is not clear for me the single cell technology used, is it STRT-seq like 5' signal or 3' UTR signal. I would like to have access to the data in the GEO link to verify all this information.

6) Why did the authors use the Hg19 assembly and then use the lift gencode annotations?

7) What is the final sequencing depth of the samples?

8) When performing the dimension reduction of the single cell data, authors show PCA and also tSNE and UMAP. In the 3 the PCA does not show clear separation. Could the authors provide the PCs and the genes involved on them? It is not clear from the PCA the different groups.

When showing the tSNE, there is a pattern that reflects the different timepoints. Could this be a reflection of the batches? I would recommend to perform a clustering on the PCA and show the tSNE. Since the authors used Seurat v3, it will be easy to call clusters. I am also concerned if this could be a reflection of the batches. Did the authors try to integrate the data and compare the different conditions?

The tSNE plots do not reflect the actual location of different cell groups, they do not give information about the position and distances. They are used only for visualization purposes. I do not find correct the technique used by the authors, on splitting the tSNE based on the position on the tSNE visualization. I would like to see as mentioned before, the PCs and their loading composition and from there the clustering based on the PCAs, with the final visualization of the clusters. The authors do also not mention which genes were used for the reduction, all the genes, variable genes... this will lead to different results.

9) In the methods section the links of the public and previously published sequencing data are missing, and also how they were processed.

10) Using the RRG seems an interesting way to see the progression of the expression of the genes in the different treatments. However, I find misleading as mentioned before, that "RRGs reflects the PCs". I want to see the PC loading and if these are significant. It is clear that there are patterns on expression through the treatments, but the way to process the single cell data clustering seems not totally correct.

11) On the results showed on figure5, I would like to see how these pairs of genes are coexpressed or not using the normalized expression used to call the PCA and the tSNE.

12) If the authors perform an unsupervised clustering how is that compared to their clusters?

Note: for these responses, line numbers refer to the marked-up version depicting the tracked changes to the manuscript. Line numbers may differ slightly in version of the manuscript without markup.

Reviewer #3 (Remarks to the Author):

Hoffman et al. study and describe the cellular heterogeneity in response to glucocorticoids in a breast cancer cell line, using a time course of Dex treatment. In the analysis they compare results from bulk and single cell RNA-seq data. In general I found some of the methodologies used not very conventional, for the analysis of single cell data. I am missing some specifications on the methodology and data processing, and also the reviewers taken to the GEO accession. The authors show that there is heterogeneity but they do not show an unsupervised clustering of the data, which if there are significant transcriptional heterogeneous groups, should be reflected easily. In order to consider the manuscript, I would ask the authors to explain the doubts raised and to answer my concerns.

1) Is usual to find differences in the detection of DEGs when comparing bulk and single cell. Lowly expressed genes will be more difficult to be identified in single cell. Also bulk RNA-seq losses resolution of the single cell because is basically an average of the expression for all the pooled cells. The authors use a method of ranking gene expressions by quantiles to be able to compare both single and RNA-seq.

When comparing bulk and single cells DEGs, authors mention that there is a high variability between replicates in the bulk RNA-seq. Authors should explain why there is such a variability, are these technical or biological replicates. For any bulk RNA-seq experiment I would expect some correlation or PCA between the replicates. High variability could mean that these can not be used as actual replicates. The model used for differential gene expression should consider a normalization of the read counts in the cases that there is a high variability of the replicates. Can the authors explain there is such variability?

The reviewer raises a critical point, but we believe that the issue here is that this section was poorly worded on our part. We fully agree that high variability between replicates would be a significant cause for concern. Our suggestion about variability between replicates here was not meant to indicate that there was global variability between our biological replicates. Rather, we intended to suggest that the small number of scRNAseq-unique DEGs with high fold changes may not pass significance thresholds in the bulk RNAseq because of higher variability between bulk replicates specifically for ZNF703. We have specifically noted this in lines 197 and 201 of this section to be clearer.

2) Authors identify ZNF703 is regulated by GR by identifying GR peaks in a region of 99Kb upstream its promoter. Then, authors verify a possible enhancer in this peak region when they investigate public data including H3K27ac. When investigating the upstream region, the authors choose a large window. If we include gencode annotations to this region, there are several lncRNAs that match exactly the position of these GR peaks. Could be that this enhancer or promoter belong to these upstream lncRNAs? Authors should compare to available Hi-C data in order to identify if there is evidence of a link between this enhancer and the promoter of ZNF703. I would like to see if there is a link between the enhancers and the promoter. This would give more strong evidence of the regulation of ZNF703 by GR. Can the authors include in the supplementary figure 3 the tracks from gencode to see any other overlapping annotated gene on the selected region?

The authors generated many sequencing data, it will be useful for the reviewers to have a genome browser link with the tracks to see the actual location of the peaks, and in this case investigate the ZNF703 region.

The reviewer is correct that the GR peaks at -97kb and -98.5kb do overlap a potential LincRNA. However, we do not believe that this precludes these GR binding sites from being involved in regulation of ZNF703. Enhancers and promoters can be promiscuous, and do not “belong” to individual genes, thus these binding sites could regulate the expression of both the LincRNA and ZNF703. Alternatively, this annotated LincRNA could represent detection of an enhancer RNA, which have been shown to be associated with chromatin interactions by steroid hormone receptors such as GR and ER. We have added the gencode gene annotations to supplemental figure 3 and indicated the presence of these potential LincRNAs the text on line 206. Furthermore, our previously published ChIP-seq data is publicly available at <https://www.ncbi.nlm.nih.gov/geo/query/acc.cgi?acc=GSE112491>.

3) When authors mention that ZNF703 was detected in more than 95% of the cells. They are counting all the cells with expression more than 0 for this gene? Or there is any specific threshold.

There is no specific threshold, we are counting all cells in which ZNF703 is detected.

4) The authors are going to deposit the data in GEO repository. It is essential that data is available for the reviewers before any decision. There for, I would like to have a access to the reviewers token.

Our data is now posted at <https://www.ncbi.nlm.nih.gov/geo/query/acc.cgi?acc=GSE141834> and our data availability statement has been updated (lines 537-539). The data is available for reviewers using the token below:

To review GEO accession GSE141834:

Go to <https://www.ncbi.nlm.nih.gov/geo/query/acc.cgi?acc=GSE141834>

Enter token ezsdeaskbfqpv into the box

5) It is not clear for me the single cell technology used, is it STRT-seq like 5' signal or 3' UTR signal. I would like to have access to the data in the GEO link to verify all this information.

We used the Illumina WTA 3' Library Prep kit and sequenced each library/timepoint to over 200 million reads. This was stated in the methods section, as excerpted here:

“The BioRad ddSeq Single-Cell Isolator was used to create single cell emulsions and the Illumina SureCell WTA 3' Library Prep kit was used to process samples and generate scRNAseq libraries. In order to detect as many genes as possible, libraries were sequenced at high depth on the Illumina Novaseq 6000 to obtain more than 200 million raw reads per library.”

6) Why did the authors use the Hg19 assembly and then use the lift gencode annotations?

For these experiments, we continued to use the hg19 assembly to match all of our previous data sets as well as several other published data sets that we were interested in comparing to our data. For gene annotations, we used v28lift37 release (provided directly by Gencode) as it was the most up-to-date comprehensive gene-set when we originally processed these data sets.

7) What is the final sequencing depth of the samples?

For the scRNAseq, we obtained over 200 million for each library. For bulk RNAseq, we obtained over 140 million reads per biological replicate.

8) When performing the dimension reduction of the single cell data, authors show PCA and also tSNE and UMAP. In the 3 the PCA does not show clear separation. Could the authors provide the PCs and the genes involved on them? It is not clear from the PCA the different groups. When showing the tSNE, there is a pattern that reflects the different timepoints. Could this be a reflection of the batches? I would recommend to perform a clustering on the PCA and show the tSNE. Since the authors used Seurat v3, it will be easy to call clusters. I am also concern if this could be a reflection of the batches. Did the authors try to integrated the data and compare the different conditions?

The tSNE plots do not reflect the actual location of different cell groups, they do not give information about the position and distances. They are use only for visualization purposes. I do not find correct the technique used by the authors, on splitting the tSNE based on the position on the tSNE visualization. I would like to see as mentioned before, the PCs and their loading composition and from there the clustering based on the PCAs, with the final visualization of the clusters. The authors do also not mention which genes were used for the reduction, all the genes, variable genes... this will lead to different results.

We would like to emphasize that the cells were processed simultaneously, and the scRNAseq libraries were generated at the same time using the same batch of reagents, so we do not believe that there are any batch effects confounding the clustering of the data set.

However, we do agree that our “slicing” of the tSNE plots was not very conventional. We used this approach because, in presenting the preliminary data to our colleagues, we found that audiences were more receptive to this analysis than the use of *de novo* clusters based on the PCAs. We agree that the use of *de novo* PCA-derived clusters is more appropriate for publication of the data. As such, we have replaced Figure 3E-F with new Figure panels. These new figures show that PCA-based cluster calling results in 7 clusters of cells. When the compositions of these clusters are examined, 3 of the 7 clusters have a majority of cells from a single timepoint while 4 of 7 clusters are more heterogeneous. All 7 clusters contained cells from at least 3 treatment timepoints. Thus, we believe that the observations and conclusions that we made in this section of the results remain sound. We have edited the last paragraph of this section (lines 256-266) as well as the legend for Figure 3 (lines 246-247) to correspond to the new figure panels.

Additionally, for PCA, we used the top 500 variable genes in the data set. We have added this information to our description of the PCA in the results section (lines 219-220)

9) In the methods section the links of the public and previously published sequencing data are missing, and also how they were processed.

We have added a link to our previous GEO submission (<https://www.ncbi.nlm.nih.gov/geo/query/acc.cgi?acc=GSE112491>) to our data availability statement.

10) Using the RRG seems an interesting way to see the progression of the expression of the genes in the different treatments. However, I find misleading as mentioned before, that “RRGs reflects the PCs”. I want to see the PC loading and if these are significant. It is clear that there are patterns on expression through the treatments, but the way to process the single cell data clustering seems not totally correct.

We are afraid that we do not fully understand this comment and apologize if we were unclear. In the manuscript, we stated that:

“Mapping the RRG data as the color of the PCA plot resulted in a strong gradient from left to right, demonstrating that PC1 was driven by the RRG (Figure 4D). Thus, the greatest source of variance within the scRNAseq dataset was the heterogeneity in the transcriptional response to Dex.”

This statement was not drawn from any other processing of the data, simply the observation that the RRG models very nicely along the axis of the first principal component. By definition, the first principal component represents the largest possible amount of variance in the data set. As such, we feel that the logical conclusion is that the primary source of variance in the scRNAseq data set is the transcriptional response to Dex.

11) On the results showed on figure5, I would like to see how these pairs of genes are coexpressed or not using the normalized expression used to call the PCA and the tSNE.

This comparison requested by the reviewer is explicitly shown in figure panels C and D, as well as supplemental figure 7. These figures depict the normalized expression of several gene pairs, with each gene shown individually and the two genes merged. In panels C we highlight two highly correlated dex-responsive genes and in figure D we show two poorly correlated dex-responsive genes. In the supplemental figure, we show 4 additional pairs of genes.

12) If the authors perform an unsupervised clustering how is that compared to their clusters?

Please see our response to concern #8

REVIEWERS' COMMENTS:

Reviewer #3 (Remarks to the Author):

All my major concerns have been appropriately addressed. This version of the manuscript has improved quality and the methods used are more correct. It is suitable for publication in Communications Biology.

Minor comments:

a) In line 260: "that" is repeated
"multiple time points, indicating that that subsets of cells are transcriptionally indistinguishable across the"

b) I will recommend to include this information, from my point 7, in the methods section;

For the scRNAseq, we obtained over 200 million for each library. For bulk RNAseq, we obtained over 140 million reads per biological replicate.

Response to Reviewers

REVIEWERS' COMMENTS:

Reviewer #3 (Remarks to the Author):

All my major concerns have been appropriately addressed. This version of the manuscript has improved quality and the methods used are more correct. It is suitable for publication in Communications Biology.

Minor comments:

a) In line 260: "that" is repeated

"multiple time points, indicating that that subsets of cells are transcriptionally indistinguishable across the"

We have corrected this error in the text

b) I will recommend to include this information, from my point 7, in the methods section;

For the scRNAseq, we obtained over 200 million for each library. For bulk RNAseq, we obtained over 140 million reads per biological replicate.

The read information for the scRNAseq was already present in the methods section (lines 502-503). We have added the read information for the bulk RNAseq to the methods on line 495.